# Tri-Compartmental Restriction Spectrum Imaging Breast Model Distinguishes Malignant Lesions from Benign Lesions and Healthy Tissue on Diffusion-Weighted Imaging

**DOI:** 10.3390/cancers14133200

**Published:** 2022-06-30

**Authors:** Alexandra H. Besser, Lauren K. Fang, Michelle W. Tong, Maren M. Sjaastad Andreassen, Haydee Ojeda-Fournier, Christopher C. Conlin, Stéphane Loubrie, Tyler M. Seibert, Michael E. Hahn, Joshua M. Kuperman, Anne M. Wallace, Anders M. Dale, Ana E. Rodríguez-Soto, Rebecca A. Rakow-Penner

**Affiliations:** 1Department of Radiology, University of California-San Diego, La Jolla, CA 92093, USA; albesser@health.ucsd.edu (A.H.B.); l1fang@health.ucsd.edu (L.K.F.); m2tong@health.ucsd.edu (M.W.T.); hojeda@health.ucsd.edu (H.O.-F.); cconlin@health.ucsd.edu (C.C.C.); sloubrie@health.ucsd.edu (S.L.); tseibert@health.ucsd.edu (T.M.S.); mehahn@health.ucsd.edu (M.E.H.); jkuperman@health.ucsd.edu (J.M.K.); amdale@health.ucsd.edu (A.M.D.); a6rodriguezsoto@health.ucsd.edu (A.E.R.-S.); 2Department of Circulation and Medical Imaging, Norwegian University of Science and Technology, Postboks 8905, 7491 Trondheim, Norway; andreassen.maren@gmail.com; 3Department of Radiation Medicine and Applied Sciences, University of California-San Diego, La Jolla, CA 92093, USA; 4Department of Bioengineering, University of California-San Diego, La Jolla, CA 92093, USA; 5Department of Surgery, University of California-San Diego, La Jolla, CA 92093, USA; amwallace@health.ucsd.edu; 6Department of Neuroscience, University of California-San Diego, La Jolla, CA 92093, USA

**Keywords:** breast diffusion MRI, breast diffusion, restriction spectrum imaging, benign breast lesions

## Abstract

**Simple Summary:**

Women at high risk for breast cancer are regularly screened using contrast-enhanced magnetic resonance imaging (MRI) to identify potential malignancy. Diffusion-weighted MRI (DW-MRI) is a non-contrast technique that measures the differential movement of water molecules in tissues and is sensitive to cancer cells. In this study, we use a multi-exponential advanced DW-MRI model called restriction spectrum imaging (RSI) to characterize the diffusion characteristics of malignant lesions, benign lesions, and healthy breast tissue to help differentiate benign from malignant disease. In a cohort of patients, we show that cancer exhibits more restricted diffusion compared to benign breast lesions and healthy tissue, whereas benign and healthy tissue are not different from each other.

**Abstract:**

Diffusion-weighted MRI (DW-MRI) offers a potential adjunct to dynamic contrast-enhanced MRI to discriminate benign from malignant breast lesions by yielding quantitative information about tissue microstructure. Multi-component modeling of the DW-MRI signal over an extended *b*-value range (up to 3000 s/mm^2^) theoretically isolates the slowly diffusing (restricted) water component in tissues. Previously, a three-component restriction spectrum imaging (RSI) model demonstrated the ability to distinguish malignant lesions from healthy breast tissue. We further evaluated the utility of this three-component model to differentiate malignant from benign lesions and healthy tissue in 12 patients with known malignancy and synchronous pathology-proven benign lesions. The signal contributions from three distinct diffusion compartments were measured to generate parametric maps corresponding to diffusivity on a voxel-wise basis. The three-component model discriminated malignant from benign and healthy tissue, particularly using the restricted diffusion *C*_1_ compartment and product of the restricted and intermediate diffusion compartments (*C*_1_ and *C*_2_). However, benign lesions and healthy tissue did not significantly differ in diffusion characteristics. Quantitative discrimination of these three tissue types (malignant, benign, and healthy) in non-pre-defined lesions may enhance the clinical utility of DW-MRI in reducing excessive biopsies and aiding in surveillance and surgical evaluation without repeated exposure to gadolinium contrast.

## 1. Introduction

Dynamic contrast-enhanced magnetic resonance imaging (DCE-MRI) demonstrates the highest sensitivity (71–94%) to detect breast cancer compared to conventional imaging (ultrasound, mammography, and tomography). DCE breast MRI evaluation is based on morphology, distribution, and kinetics; these features are incorporated in the assessment of malignancy potential by the Breast Imaging Reporting and Data System (BI-RADS) [1,2,3,4,5]. The American College of Radiology Commission on Breast Imaging appropriateness criteria recommend that women at high risk for breast cancer begin annual screening with DCE-MRI as early as 25 years old [6]. However, DCE-MRI is associated with limitations including variable specificity rate (61–97%), which leads to excessive biopsies and increases patient anxiety [1,2,3,7]. 

Diffusion-weighted MRI (DW-MRI) has emerged as a method with the potential to differentiate benign from malignant breast lesions and yields quantitative information about tissue microstructure without the use of exogenous contrast [8,9]. Information about tissue structure can be provided by measuring an apparent diffusion coefficient (ADC) in varying degrees of diffusion weighting (*b*-values), due to the differential Brownian diffusion of water molecules [8]. Although DW-MRI is part of the standard of care for many body applications and included in the final assessment for detection of malignancies in the liver, prostate, and ovaries, there continue to be limitations and no lexicon descriptors or quantification of DW-MRI and ADC in BI-RADS at this time. These include limitations due to significant geometric distortions which can impair the evaluation of small lesions [9], variations in ADC values related to acquisition parameters across sites [10,11,12], and substantial overlap in ADC values for malignant and benign lesions [13,14]. Therefore, the clinical utility of using ADC as a robust imaging biomarker of the probability of malignancy of lesions is limited and mostly recommended as an adjunct measurement with DCE-MRI [15,16]. To advance DW-MRI as a reliable non-contrast adjunct to DCE-MRI in evaluating breast lesions, a more comprehensive diffusion model is needed that more accurately describes the complex microenvironment of the breast, consisting of healthy fat and fibroglandular tissues, benign lesions, and malignant lesions.

Multi-component modeling of the DW-MRI signal over extended ranges of *b*-values (typically up to 2000–3000 s/mm^2^) theoretically isolates the slowly diffusing (restricted) water component in tissues [8]. The magnitude of this restricted diffusion component is hypothesized to be modulated by both cellularity and nuclear volume fraction of individual cells [17]. In the restriction spectrum imaging (RSI) model, each component is defined by a constant apparent diffusion coefficient representing a distinct pool of water diffusion signal [18]. The main outputs of this technique are maps of the relative size of each component. In brain [19], prostate [20,21], and breast [22,23,24], RSI has demonstrated potential to distinguish cancer tissues from non-cancer tissues by isolating this slowly diffusing water signal, and corresponding magnitude. Vidić et al. demonstrated that the normalized magnitude of the slowest element in a two-component model performed well in discriminating signal intensity within pre-defined benign and malignant breast lesion regions of interest (ROIs) [22].

A further optimized three-component RSI model to fit the DW-MRI signal in the breast demonstrated the ability to differentiate malignant from healthy fibroglandular tissue [23], with comparable contrast-to-noise ratio and area-under-the-curve to conventional DCE-MRI [23,24]. This three-component model was used to generate a classification algorithm for non-predefined lesions which demonstrated the clinical utility of RSI in breast cancer diagnosis [24]. However, the study did not investigate the model’s ability to discriminate malignant and benign breast lesions. Further refining this model to include discrimination of benign lesions from both malignant lesions and healthy tissue would increase the clinical robustness of the model and increase the radiologist’s confidence in ascribing a lesion as malignant or benign. Therefore, the aim of this study was to evaluate the utility of the three-component RSI model to distinguish between malignant, benign, and healthy tissue in a small patient population with biopsy-proven malignant and benign breast lesions.

## 2. Materials and Methods

### 2.1. Subjects

A retrospective study of the RSI characteristics of benign and malignant breast lesions was conducted. This study was approved by the Institutional Review Board of the authors’ institution. All patients were females, 20–78 years old. Patients with known breast malignancy and who were imaged with DCE-MRI and the relevant diffusion protocol at our single institution between July 2015 and December 2019 were identified (208 total patients) (Figure 1). 

Patients with both pathology-proven malignancy and one or more synchronous pathology-proven benign lesion(s) in either the contralateral or ipsilateral breast were included (122 patients) to control for interpatient variability. Therefore, 79 patients without synchronous benign lesions were excluded. Patients previously treated with a cytotoxic regimen, chemotherapy, or ipsilateral radiation therapy prior to MRI scanning were excluded from analysis (86 patients). Additionally, patients with simple cysts that are readily identified with standard imaging (11 patients), non-pathology-proven benign lesions (8 patients), and incomplete or alternative MRI protocol (12 patients) were excluded from the analysis. Therefore, our final patient population consisted of 12 patients with concurrent pathology-proven malignant and benign lesions in either breast.

### 2.2. Histopathology

Lesions were pathologically assessed via core needle or excisional biopsy. All 12 patients had only one malignant lesion and at least one benign lesion, with 2 of these patients having an additional ipsilateral benign lesion for a total of 14 benign lesions (Table 1). All of the 14 total benign lesions were pathology-proven benign and included stromal fibrosis, fibroadenomas, focal ductal hyperplasia, benign sclerosing adenosis, radial scar, and benign breast parenchyma. The size of the benign lesions ranged from 0.5 × 0.5 cm to 7.1 × 1.4 cm based on the radiologists’ reports.

Malignant lesions included invasive ductal carcinoma (IDC), invasive mammary carcinoma (IMC), invasive lobular carcinoma (ILC), ductal carcinoma in situ (DCIS), IDC + DCIS, and IMC + DCIS. Lesions ranged from 2.1 × 1.9 cm to 6.2 × 5.8 cm based on the radiologists’ reports and represented masses, non-mass enhancement (NME), and mass + NME (Table 1).

### 2.3. MRI Data Acquisition

MRI data were collected using a 3.0T scanner (MR750, DV25-26, GE Healthcare, Milwaukee, WI, USA) with an 8-channel breast array coil. DCE-MRI images were acquired following gadolinium administration, and contrast kinetic graphs were generated using CADstream (Merge Healthcare Inc., Chicago, IL, USA) software. Images were reported according to BI-RADS recommendations [25]. In addition to T_1_- and T_2_-weighted images, axial reduced field-of-view (FOV) echo-planar imaging DW-MRI was performed with the following parameters: spectral attenuated inversion recovery (SPAIR) fat suppression, echo time (TE) = 82 ms, repetition time (TR) = 9000 ms, *b*-values (number of diffusion directions) = 0, 500 (6), 1500 (6), and 4000 (15) s/mm^2^, gradient pulse duration (δ) = 29.34 ms, gradient pulse time interval (Δ) = 37.67 ms, FOV = 160 × 320 mm^2^, acquisition matrix = 48 × 96, reconstruction matrix = 128 × 128, voxel size = 2.5 × 2.5 × 5.0 mm^3^, phase-encoding (PE) direction anterior-posterior (A/P), and no parallel imaging. The *b* = 0 s/mm^2^ volumes were collected in the A/P and posterior–anterior (P/A) PE directions to correct DW-MRI data for geometric and intensity distortions due to *B_0_* inhomogeneities using the reverse polarity gradient (RPG) method [26,27]. For the diffusion sequence, δ and Δ time constants were fixed, while the gradient magnitude was manipulated to produce different *b*-values.

### 2.4. Data Processing

Data were processed and analyzed using MATLAB R2016b (The MathWorks Inc., Natick, MA, USA), as previously described in Rodríguez-Soto et al. [23]. Briefly, after distortion correction with RPG, datasets were noise corrected by estimating the noise floor as the average of the background signal determined from a histogram of the DW-MRI data. The noise floor value was then subtracted from all voxels. Following noise correction, data were corrected for eddy current artifacts, then all diffusion directions at a given *b*-value were averaged. Averaged DW-MRI datasets were normalized by the 98th percentile of the *b* = 0 s/mm^2^ volume to preserve T_2_ information while mitigating the chance of using an arbitrarily high noise value for normalization, as done previously [23].

Previously, Rodríguez-Soto et al. established that the diffusion signal, *S_diff_*, in breast cancer is best modeled as a linear combination of three exponential decays as described by the RSI framework [18,23]:(1)$$S_{diff}\left(b\right)=C_{1}+C_{2}e^{-b\cdot 1.5\times 10^{-3}}+C_{3}e^{-b\cdot 10.8\times 10^{-3}}$$
where *C_i_* is the signal contribution of each exponential component (*i* = 1, 2, and 3 for the three-component breast model) and *b* represents the *b*-values. In brief, Rodríguez-Soto et al. first estimated diffusion coefficients of each exponential component (*D*_1_ = 0 mm^2^/s (restricted diffusion), *D*_2_ = 1.5 × 10^−3^ mm^2^/s (hindered diffusion), and *D*_3_ = 10.8 × 10^−3^ mm^2^/s (pseudo-diffusion)) through global fitting of the tri-exponential model to cancer and control tissue. To estimate the parametric *C_i_* maps, these diffusion coefficients are fixed to enable comparison of signal contributions across different voxels and patients, and a non-negative least squares fitting of the model to the signal versus *b*-value curve from each voxel is performed [23]. 

The resulting *C*_1_, *C*_2_, and *C*_3_ maps were directly estimated from the model. Additionally, *C*_1_*C*_2_, the product of the *C*_1_ and *C*_2_ components, was demonstrated by Andreassen et al. to discriminate between cancer and healthy breast tissue in a clinically helpful manner and was thus included in our analysis [24]. In the present work, the square root of the product, $\sqrt{C_{1}C_{2}}$, was also estimated to provide the information of *C*_1_*C*_2_ at the same scale as the RSI *C_i_* map outputs. Additionally, conventional mono-exponential ADC maps were computed using averaged DW-MRI datasets without noise correction, using *b*-values up to 1500 s/mm^2^. Negative and undefined ADC values were excluded.

Using all available data in the exam protocol (including DCE-MRI and anatomical T_2_-weighted images), full volume regions of interest (ROIs) were manually drawn on DW-MRI images for all malignant and benign lesions, as shown in Figure 2. 

Malignant ROIs were drawn for the lesions corresponding to pathology-proven cancer, and benign ROIs were drawn for those corresponding to pathology-proven benign lesions. Healthy control ROIs containing healthy fat and fibroglandular tissue were defined as boxes of 33,600 voxels placed in the contralateral breast relative to the malignancy being evaluated, drawn on DCE-MRI images. Healthy control ROIs were then resampled to the DW-MRI resolution to cover 500 voxels. Any benign lesion voxels contained within healthy control ROIs were excluded. Additionally, the axillary region, large cysts (>2.5 cm), and susceptibility artifacts (e.g., from surgical clips) were excluded. ROIs were reviewed and approved by a breast radiologist (RRP). After assessing data normality, the median of each ROI was computed, resulting in a single data point per tissue type per patient.

### 2.5. Statistical Analysis

Statistical analyses were performed using R statistical programming (R Foundation for Statistical Computing, Vienna, Austria). Shapiro–Wilk test for normality was used to evaluate the normality of data within each ROI. Levene’s test for homogeneity of variance was used to examine the normality of data across patients, followed by ranked two-way repeated-measures analysis of variance (ANOVA) to evaluate the effects of tissue type and diffusion components. Individual differences were evaluated by post hoc Wilcoxon signed-rank test with Bonferroni correction to preserve rigor. The threshold for significance (α) was set at 0.05 for all analyses. Results are reported as median and interquartile range values.

## 3. Results

### 3.1. Contrast Enhancement Kinetics of Benign Lesions Suggest Suspicious Pathology

Pre-biopsy MRI reports were reviewed for all benign lesions in the study group. All benign lesions were interpreted as suspicious and required biopsy (Figure 3). These lesions displayed either Type II or Type III contrast enhancement pattern, thus increasing suspicion for malignancy per BI-RADS criteria. [1,2,3,4,5] (Figure 3).

### 3.2. Malignant Lesions Display Higher C_1_ and C_2_ Compartment Values Compared to Benign Lesions and Healthy Tissue

On average, malignant lesion ROIs contained approximately 20-fold the number of voxels compared to benign lesion ROIs (Table 2). A representative example of malignant, benign, and healthy tissue ROIs on both DW-MRI *b* = 0 s/mm^2^ images and peak intensity post-contrast images in a single patient are shown in Figure 3.

Visually, the malignant, benign, and healthy tissue displayed different signal intensities across compartment maps (Figure 4). For example, malignant lesions (Figure 4b,c,e,f) display the highest signal in *C*_1_, *C*_2_, *C*_1_*C*_2_, and $\sqrt{C_{1}C_{2}}$ compared to benign lesions (Figure 4h,i,k,l) and healthy tissue (Figure 4n,o,q,r). 

Malignant lesions had the highest signal contributions in the *C*_1_ (*p* < 0.001) compartment compared to benign lesions and healthy tissue, with median (interquartile range) values of 0.32 (0.18) for malignancy, 0.05 (0.12) for benign tissue, and 0.08 (0.13) for healthy tissue, respectively (Figure 5, Table 2 and Table 3). In the *C*_2_ compartment, benign lesions were not different from malignant lesions (*p* = 0.30) or healthy tissue (*p* = 0.90), but malignant lesions had higher (*p* < 0.01) *C*_2_ values compared to healthy tissue. The *C*_2_ values were 2.6 (1.7) for malignant lesions, 1.7 (1.5) for benign lesions, and 0.90 (0.73) for healthy tissue. The *C*_3_ signal was comparable (*p* = 1) for all tissue types: 0.13 (0.40) for malignancy, 0.34 (0.70) for benign tissue, and 0.34 (0.43) for healthy tissue. Benign lesions were not significantly different from healthy tissue in any of the compartments (Figure 5, Table 3). Additionally, mono-exponential ADC did not significantly differ between the three tissue types (Table 2).

### 3.3. Combinations of C_1_ and C_2_ Also Discriminate Malignant and Benign Lesions

Overall, *C*_1_*C*_2_ and $\sqrt{C_{1}C_{2}}$ combination maps resulted in significantly higher (*p* < 0.05) values for malignant lesions compared to benign lesions and healthy tissue (Table 2, Figure 5). The *C*_1_*C*_2_ median values were 0.7 (1.0), 0.08 (0.29), and 0.03 (0.05) for malignant lesions, benign lesions, and healthy tissue, respectively. The product *C*_1_*C*_2_ discriminated malignant lesions from benign (*p* < 0.01) and healthy (*p* < 0.001) tissues (Table 3). Additionally, the square root of the product $\sqrt{C_{1}C_{2}}$ separated malignant from benign (*p* < 0.01) and healthy (*p* < 0.01) tissues (Table 3). The values for $\sqrt{C_{1}C_{2}}$ were 0.81 (0.58), 0.29 (0.46), and 0.16 (0.16) for malignant, benign, and healthy tissue, respectively. However, neither *C*_1_*C*_2_ nor $\sqrt{C_{1}C_{2}}$ significantly distinguished between benign lesions and healthy tissue.

## 4. Discussion

The main objective of this study was to evaluate how the breast-optimized three-component RSI model can discriminate malignant lesions, benign lesions, and healthy breast tissue. We sought to understand the limitations of this technique and assess the degree of overlap between malignant, benign, and healthy breast tissue to evaluate the potential clinical utility of the model in decreasing the number of excessive biopsies.

Our results show that a three-component advanced DW-MRI model can discriminate malignant tissue from benign lesions and healthy breast tissue in a small group of patients. Additionally, the product and the square root of the product of two of the compartments (*C*_1_ and *C*_2_) perform similarly to *C*_1_ alone in discriminating malignant lesions from benign lesions and healthy tissue. Importantly, we found that benign lesions were not significantly different from healthy breast tissue in any compartment or compartment combination.

Understanding what differences comprise these three groups may increase our understanding of the diffusion metric results. The three compartments theoretically represent different degrees of diffusion, where *C*_1_ is hypothesized to represent the most restricted diffusion derived from cancer cells or the restricted water component within adipocytes in fatty tissue [28], *C*_2_ is hypothesized to represent intermediate or hindered diffusion predominantly from fibroglandular tissue, and *C*_3_ is hypothesized to represent the least restricted diffusion such as pools of fluid or flow through blood vessels [23,24]. Our data suggest that malignant lesions display high conspicuity in the *C*_1_ compartment, both in the individual *C*_1_ map and in the context of a derived *C*_1_*C*_2_ or $\sqrt{C_{1}C_{2}}$ parameter. In other words, *C*_1_ can be used to distinguish the malignancy of a lesion due to increased restricted diffusion compared to benign lesions and healthy tissue. Though benign lesions had a generally higher *C*_2_ compartment than healthy tissue, there was considerable variability between samples in this compartment, resulting in nonsignificant differences between malignant and benign tissues and benign and healthy tissues. Further, all three tissue types demonstrated comparable degrees of unhindered diffusion, represented by the *C*_3_ compartment, suggesting that vascular flow was similar across tissue types. Overall, the compartment demonstrating the most restricted diffusion separated malignant from benign lesions, regardless of lesion subtype and size.

Interestingly, we found no significant differences between benign lesions and healthy breast tissue in any of the compartments. The three-compartment RSI model used in this study follows that used in Andreassen et al. and Rodríguez-Soto et al., where control ROIs comprised the entire contralateral breast, including any benign lesions, were used to establish the model [23,24]. Our findings support their assumption that benign lesions generally display similar diffusion properties as healthy breast tissue in a model where the diffusion signal is decomposed into three distinct compartments. Only a few studies have evaluated the ability of DW-MRI in discriminating healthy and benign breast tissues, with mixed results: Sharma et al. [29] found significantly higher ADC values for healthy tissues compared to benign lesions, while Woodhams et al. [14] demonstrated no significant differences between the tissue types. Since the ADC of healthy breast tissue has been found to vary with breast density [30], our study was limited to subjects with concurrent benign and malignant lesions, with the healthy control also derived from the same patient. This allowed us to perform a paired analysis to mediate the potential effects of breast density on diffusion metrics.

DW-MRI is a potential complement to DCE-MRI that may reduce the number of excessive biopsies from false-positive interpretations. In our cohort, all benign lesions were classified as suspicious from contrast enhancement kinetics and thus required tissue biopsy. Several groups have already demonstrated the ability of the mono-exponential ADC diffusion estimate in discriminating between benign and malignant lesions in the breast [22,31,32]. However, the translation of DW-MRI into clinical practice has been limited by a lack of consensus in cut-off values from variation due to geometric distortions [9], chosen sequence parameters [12], and vendor- and system-specific bias [10]. Additionally, in the literature, reported absolute ADC threshold values are in the range 0.55 × 10^−3^–1.88 × 10^−3^ mm^2^/s for malignant lesions and 1.10 × 10^−3^–2.06 × 10^−3^ mm^2^/s for benign lesions [33]. In our small cohort, we found that malignant lesions, benign lesions, and healthy breast tissue had ADC values that overlapped considerably. The wide overlap between ADC measurements for benign and malignant lesions makes ADC less robust of an imaging biomarker on a per-patient basis, and only a handful of studies have been conducted regarding separating benign lesions from healthy breast tissue [14,29]. This study extends the work by Andreassen et al. in demonstrating the potential clinical utility of an advanced DW-MRI framework that compartmentalizes diffusion on a voxel-level and is specific to the breast [24]. In a small sample of patients, the three-component RSI model was able to discriminate malignant lesions from benign lesions and healthy breast tissue.

Small sample size is the primary limitation of our study. We limited our sample subset to patients who had both concurrent biopsy-proven malignancies and biopsy-proven benign lesions from a single institution. While this aided in reducing interpatient variability, this significantly limited the number of patients eligible for evaluation. Additionally, malignant lesions were enriched in our population given that our study group consisted of women with known malignancy. Future work will also focus on prospectively analyzing women at high risk and average risk of breast cancer to further evaluate the robustness of this model as it pertains to the general population. While our data is statistically significant in this small patient sample, continued evaluation in a larger population across several institutions may lead to further optimization of the model, and this may include evaluation of even higher *b*-values (over 4000 s/mm^2^) to better discriminate the restricted diffusion compartment from intermediate and free diffusion, at the cost of increased scan time. Large components of fat in breast tissue may also require further optimization in this model, which was initially developed in less fatty tissues (e.g., brain and prostate) [8,18]. Though DW-MRI data was fat-suppressed using SPAIR, incomplete suppression may affect the derived coefficients in our small sample set. Additionally, water trapped within adipocytes and lipids may produce artificially restricted signal within fat. It is also possible that our ROIs may contain more than one tissue type (e.g., small benign lesions containing some fatty tissue, or healthy tissue ROIs containing occult malignancy). Lastly, an assumption of the RSI model is that diffusion within each voxel is isotropic, which is a gross approximation of the diffusion process. Future iterations of the model will consider anisotropic diffusion such as fractional anisotropy to address the diffusion directionality within breast tissue.

## 5. Conclusions

This three-component RSI model, along with T_2_-weighted information, provides a promising supplement to DCE-MRI for improved specificity and may play a role in the development of future non-contrast breast MRI protocols. Institution-standardized DW-MRI sequences are already included in breast imaging MRI protocols and their limitations have been briefly described in this paper. Our findings indicate that malignant lesions demonstrate high conspicuity on compartment maps indicating the most restricted diffusion compared to benign lesions and healthy breast tissue, which showed no significant differences. The model may increase radiologists’ confidence in ascribing a lesion as malignant versus benign. Future directions include applying this technique to a larger subset of patients, evaluating early high-risk atypical benign lesions and DCIS, and investigating the model’s ability to differentiate lesions of various molecular subtypes.

## Figures and Tables

**Figure 1 cancers-14-03200-f001:**
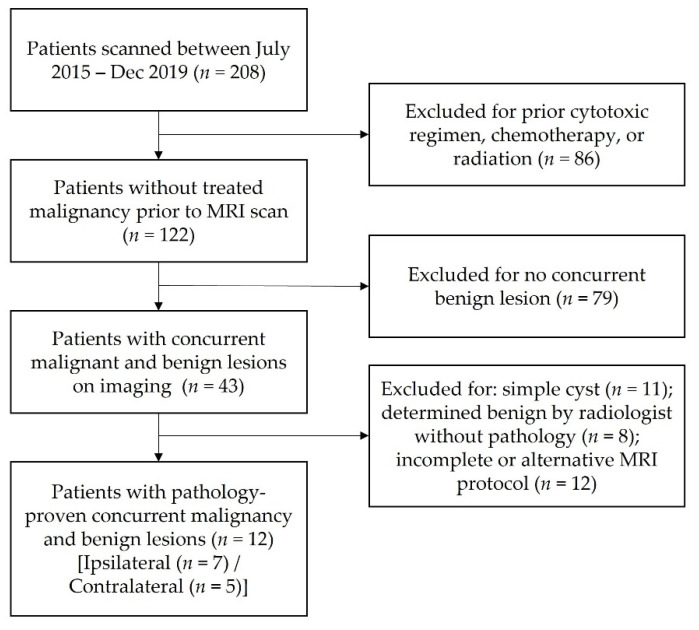
Patient selection flow chart.

**Figure 2 cancers-14-03200-f002:**
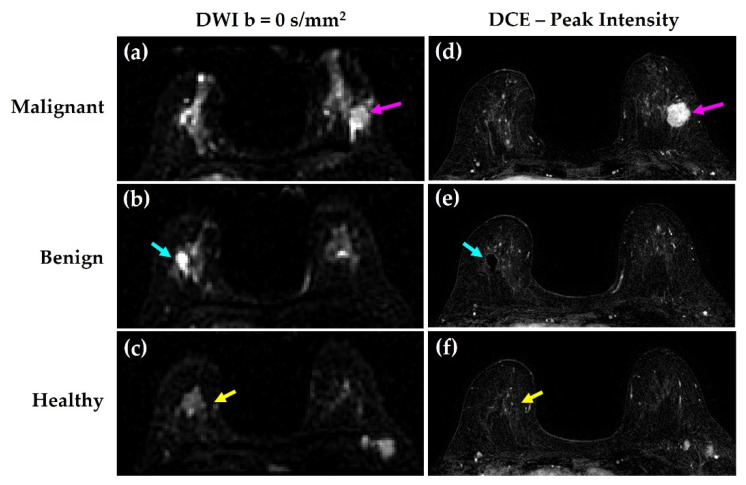
Representative lesions in one patient. Malignant (magenta), benign (cyan), and healthy (yellow) tissues are shown on (**a**–**c**) DW-MRI *b* = 0 s/mm^2^ and (**d**–**f**) DCE-MRI peak intensity images.

**Figure 3 cancers-14-03200-f003:**
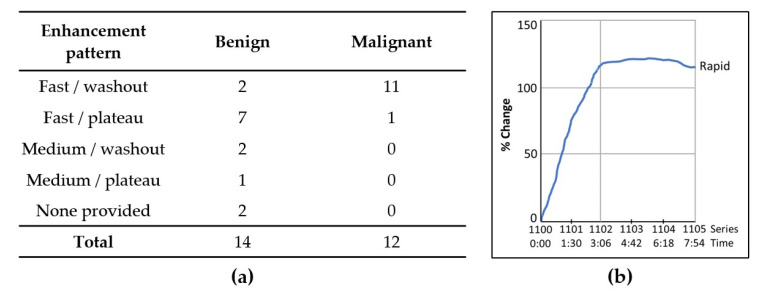
Enhancement pattern of benign and malignant lesions. (**a**) Enhancement patters are defined as: “Fast/washout” (fast initial enhancement with washout on delayed sequences); “Fast/plateau” (fast initial enhancement with plateau on delayed sequences); “Medium/washout” (medium initial enhancement with washout on delayed sequences); “Medium/plateau” (medium initial enhancement with washout on delayed sequences); “None provided” (no kinetic assessment was provided in the report). (**b**) Representative graph of a benign lesion showing fast initial enhancement with plateau on delayed sequences.

**Figure 4 cancers-14-03200-f004:**
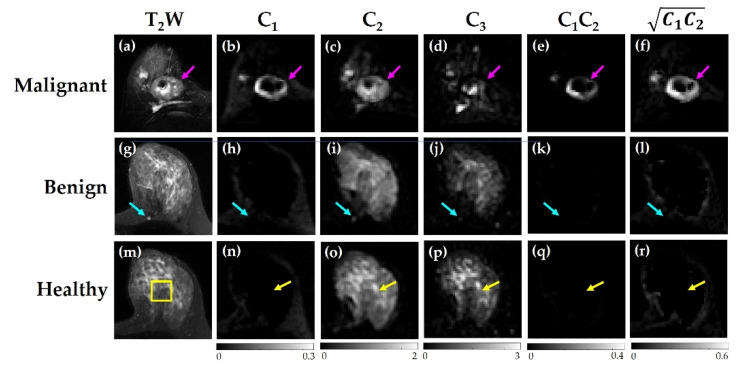
Representative maps for each compartment (*C*_1_*, C*_2_*, C*_3_*, C*_1_*C*_2_*,*
$\sqrt{C_{1}C_{2}}$*)*, with (**a**–**f**) malignant lesions, (**g**–**l**) benign lesions, and (**m**–**r**) healthy tissue ROIs indicated.

**Figure 5 cancers-14-03200-f005:**
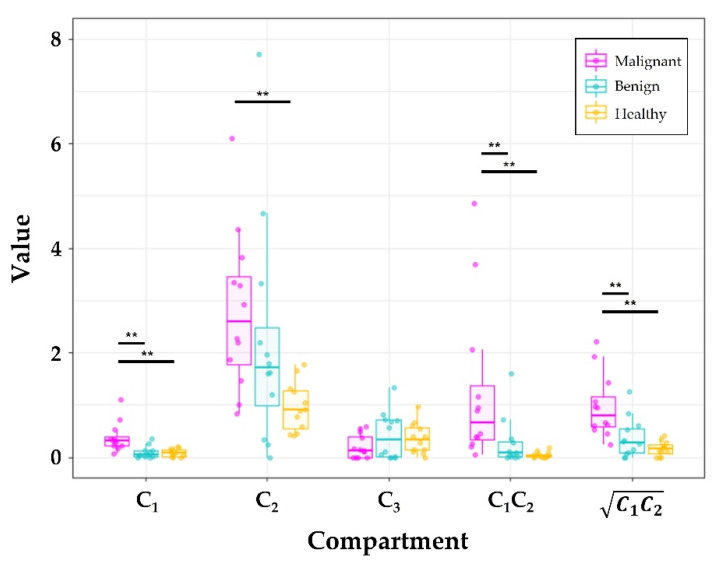
Comparison of tissue types per compartment *C*_1_, *C*_2_, *C*_3_*, C*_1_*C*_2_, and Magenta, cyan, and yellow boxes indicate malignant, benign, and healthy breast tissues, respectively. Significance from ranked two-way repeated measures ANOVA are indicated by the black bars and asterisks (** *p* < 0.001).

**Table 1 cancers-14-03200-t001:** Lesion histology.

Malignant (*n* = 12)		Benign (*n* = 14)	
**Histology**	** *n* **	**Histology**	** *n* **
IDC ^1^	7	Stromal fibrosis	6
IMC ^2^	1	Fibroadenoma	4
ILC ^3^	1	Focal ductal hyperplasia	1
IDC + DCIS ^1,4^	1	Benign sclerosing adenosis	1
IMC + DCIS ^2,4^	1	Radial scar	1
DCIS ^4^	1	Benign breast parenchyma	1
**Type**	** *n* **	**Type**	** *n* **
Mass	9	Mass	10
NME ^5^	1	NME ^5^	3
Mass + NME ^5^	2	Mass + NME ^5^	1

Abbreviations: ^1^ Invasive ductal carcinoma; ^2^ invasive mammary carcinoma, ^3^ invasive lobular carcinoma; ^4^ ductal carcinoma in situ; ^5^ non-mass enhancement.

**Table 2 cancers-14-03200-t002:** Summary of ROI and compartment map values. Data are reported as median (interquartile range). Ranked two-way repeated measures ANOVA results are indicated in the last column and row (across compartment and tissue type, respectively) (ns: *p* > 0.05).

Tissue Type	ROI Volume [cm^3^]	ADC × 10^−3^ [mm^2^/s]	*C* _1_	*C* _2_	*C* _3_	*C* _1_ *C* _2_	$\sqrt{C_{1}C_{2}}$	*p*
Malignant	6.4 (10.5)	0.94 (0.23)	0.32 (0.18)	2.6 (1.7)	0.13 (0.40)	0.70 (1.0)	0.82 (0.58)	3.5 × 10^−5^
Benign	0.5 (0.8)	1.16 (0.27)	0.05 (0.12)	1.6 (1.5)	0.34 (0.70)	0.08 (0.29)	0.29 (0.46)	0.003
Healthy	199.2 (10.5)	0.97 (0.25)	0.08 (0.13)	0.90 (0.73)	0.34 (0.43)	0.03 (0.05)	0.16 (0.16)	3.3 × 10^−4^
** *p* **		ns	1.4 × 10^−5^	0.001	ns	6.2 × 10^−6^	5.9 × 10^−6^	

**Table 3 cancers-14-03200-t003:** Pairwise comparisons of tissue type per compartment (ns: *p* > 0.05; **: *p* < 0.01).

Compartment	Groups	Bonferroni-Adjusted *p*-Value	Significance
*C* _1_	Malignant vs. Benign	0.001	**
	Malignant vs. Healthy	0.001	**
	Benign vs. Healthy	1.0	ns
*C* _2_	Malignant vs. Benign	0.30	ns
	Malignant vs. Healthy	0.01	**
	Benign vs. Healthy	0.90	ns
*C* _3_	Malignant vs. Benign	1.0	ns
	Malignant vs. Healthy	0.70	ns
	Benign vs. Healthy	1.0	ns
*C* _1_ *C* _2_	Malignant vs. Benign	0.004	**
	Malignant vs. Healthy	0.001	**
	Benign vs. Healthy	0.11	ns
$\sqrt{C_{1}C_{2}}$	Malignant vs. Benign	0.003	**
	Malignant vs. Healthy	0.008	**
	Benign vs. Healthy	0.23	ns

## Data Availability

The data presented in this study is available on request from the corresponding author.

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
