# Peer review of "Tri-Compartmental Restriction Spectrum Imaging Breast Model Distinguishes Malignant Lesions from Benign Lesions and Healthy Tissue on Diffusion-Weighted Imaging"

_cancers, 2022, doi:10.3390/cancers14133200_

Round 1
Reviewer 1 Report
This manuscript tested the described tri-compartmental imaging model using three groups of tissues (normal, benign, and malignant tissue). The continuous process of diffusion is modeled by three layers of diffusion with three constant diffusion coefficients (including zero mm^2/s instantaneous transfer), which seems an oversimplification. However, there is a critical need to develop a good diagnostic imaging tool and the authors presented a promising method. The entire description is like a case report using a fixed procedure using a good set of precious human-subjects information.
Major
· Three components: there is no basis to evaluate whether three compartments are better than two or four compartments.
· Choice of D1, D2, and D3: The rationale for choosing the particular values for D1 to D3 needs to be stated. It is recommended to show the sensitivity and dependence of the results on the selected values.
· S diff: S diff is a function of b, but how S is used to estimate C1 to C3 is not described. Please add the description of the use of S and the estimation of C1 to C3.
· Non-uniformity, size, and geometry: The diffusion process is likely to be affected by 3D direction, tumor size, and geometry of malignant/benign tissues. The described method is a gross approximation of the diffusion process. The authors are recommended to discuss the limitations.
· Time: the description does not explicitly include time as a variable. Please describe the use of b in the proposed method and its link to time.
· Validation: it is difficult to evaluate whether the described approach is successful. Please state how to validate the procedure, such as a leave-one-out method.
Reviewer 2 Report
In this paper,the authors aim to evaluate how the breast-optimized three- component RSI model can discriminate malignant lesions, benign lesions, and healthy breast tissue .
1. It is recommended to expand the sample size of patients. The sample size of this study is too small, and only 12 patients were analyzed;
2. It is recommended that a prospective study can be added;
3. In this study, the authors hope to identify benign ones through this technology so as to reduce doctors' judgment of benign as suspicious and recommend puncture. Is there a difference in the price of this technology compared to puncture? Is the time to use this technology and the threshold for imaging doctors easy for clinical promotion?
4. According to the authors, there is a significant difference between the malignant, benign and healthy values. Is that possible to calculate the cut-off value that can be used clinically (over this value can be judged as benign or healthy);
5. The axillary lymph nodes related detect are very important for patients with breast disease, whether the same method can be extended to the axilla lymph nodes?
Round 2
Reviewer 1 Report
The authors responded satisfactorily to the comments. Thank you.
Reviewer 2 Report
This article can be accepted.